# Integration of parallel pathways for flight control in a hawkmoth reflects prevalence and relevance of natural visual cues

Ronja Bigge[1,2], Rebecca Grittner[1], Anna Lisa Stöckl[1,2]*

[1]Behavioral Physiology and Sociobiology (Zoology II), University of Würzburg, Biozentrum am Hubland, Würzburg, Germany; [2]Department of Biology, University of Konstanz, Konstanz, Germany

## eLife Assessment

This **important** study investigates how hummingbird hawkmoths integrate stimuli from across their visual field to guide flight behaviour. Cue conflict experiments provide **solid** evidence for an integration hierarchy within the visual field: hawkmoths prioritise the avoidance of dorsal visual stimuli, potentially to avoid crashing into foliage, while they use ventrolateral optic flow to guide flight control. These findings will be of broad interest to enthusiasts of visual neuroscience and flight behavior.

**\*For correspondence:**
Anna.Stoeckl@uni-konstanz.de

**Abstract** An animal's behaviour is the result of multiple neural pathways acting in parallel, receiving information across and within sensory modalities at the same time. How these pathways are integrated, particularly when their individual outputs are in conflict, is key to understanding complex natural behaviours. We investigated this question in the visually guided flight of the hummingbird hawkmoth *Macroglossum stellatarum*. These insects were recently shown to partition their visual field, using ventrolateral optic flow cues to guide their flight like most insects, while the same stimuli in the dorsal visual field evoke a novel directional response. Using behavioural experiments which set the two pathways into conflict, we tested whether and how the ventrolateral and dorsal pathway integrate to guide hawkmoth flight. Combined with environmental imaging, we demonstrate that the partitioning of the visual field followed the prevalence of visual cues in the hawkmoths' natural habitats, while the integration hierarchy of the two pathways matched the relevance of these cues for the animals' flight safety, rather than their magnitude in the experimental setup or in natural habitats. These results provide new mechanistic insights into the vision-based flight control of insects and link these to their natural context. We anticipate our findings to be the starting point for comparative investigations into parallel pathways for flight guidance in insects from differently structured natural habitats.

## Introduction

An animal's behaviour is rarely guided just by one neural pathway or sensory modality. More often, parallel input from multiple sensory modalities supports various control systems at any given time: an insect might be following an odour plume, for example, while guiding its flight safely using visual and mechanosensory cues (*Duistermars and Frye, 2008*; *Vickers and Baker, 1994*). Even within the same modality, multiple pathways might act in parallel to shape the same behavioural output (*Fox and Frye, 2014*; *Frighetto and Frye, 2023*; *Nassi and Callaway, 2009*). How these parallel pathways are integrated, particularly when their individual outputs are in conflict, is key to understanding the

complex natural behaviour of animals (*Doudlah et al., 2022*). This question can be particularly well investigated in a context where the same visual stimuli evoke different responses in different parts of the visual field. Such visual field partitioning is widespread among animals and likely shaped by the frequency at which these stimuli occur in the natural habitats of the animals (*Baden et al., 2013*; *Hornstein et al., 2000*; *Turner et al., 2019*; *Alexander et al., 2022*).

Insect flight control is a behaviour that strongly depends on visual information sampled across a wide receptive field. It utilises translational optic flow, the perceived relative motion of the environment generated by the animals' own motion (*Koenderink, 1986*), to regulate the distance to the surrounding environment, and the flight speed across a wide range of insects (*Baird et al., 2005*; *Baird et al., 2010*; *David, 1982*; *Fry et al., 2009*; *Portelli et al., 2011*; *Willis and Card, 1990*; *Dyhr and Higgins, 2010*; *Kennedy and Marsh, 1974*; *Kern et al., 2012*; *Kirchner and Srinivasan, 1989*; *Kuenen and Baker, 1982*; *Portelli et al., 2010*; *Grittner et al., 2022*; *Stöckl et al., 2019*). This regulation is thought to be achieved by keeping the perceived rate of translational optic flow at a constant level across both eyes and over time (*Portelli et al., 2011*; *Franceschini et al., 2007*). In consequence, when insects encounter features that change contrast in the direction of travel, and thus generate translational optic flow, such as an array of trees or a vertically patterned wall in a flight tunnel, they slow down and keep a greater distance. Thereby, they reduce the perceived magnitude of translational optic flow (*Collett, 2002*; *Egelhaaf, 2023*; *Srinivasan et al., 1999*). Optic flow cues are also used to guide flight straightness, particularly when presented in the ventral visual field (*Bigge et al., 2021*; *Linander et al., 2017*). It is thought that insects measure optic flow across their entire visual field and respond to the area with the highest magnitude of translational optic flow within (*Portelli et al., 2011*; *Linander et al., 2015*). While this assumption has rarely been tested with dorsal optic flow cues, in honeybees and fruit flies, dorsal optic flow responses similar to the ventrolateral ones have been described (*Portelli et al., 2011*; *Mazo and Theobald, 2014*). While fruit flies weigh ventral

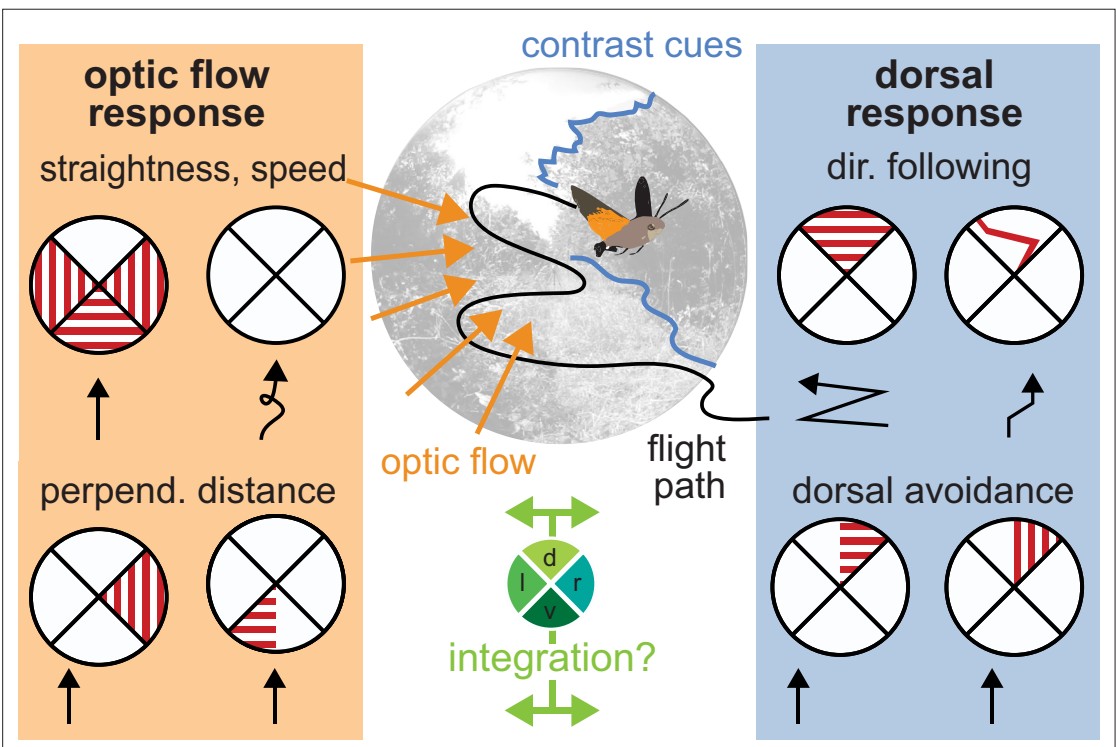

**Figure 1.** Optic flow and directional responses partition the visual field for flight control in the hummingbird hawkmoth. Flight control in most insects is strongly based on optic flow, the apparent movement of the environment across the visual field, induced by the animals' own movement. Ventrolateral translational optic flow supports straight flight, compared to featureless environments. Most insects keep the magnitude of translational optic flow constant across their eyes by adjusting their speed and perpendicular distance to optic flow-inducing textures. In hawkmoths, optic flow cues presented in the dorsal visual field induce directional responses, which align the hawkmoths' flight with the main direction of the visual cue. Moreover, hawkmoths avoid any structures in the dorsal, but not the ventral, visual field, even if they generate only weak translational optic flow.

translational optic flow stronger than dorsal optic flow, the most extreme partitioning of the visual field in terms of optic flow-based flight control has been observed in hawkmoths (*Bigge et al., 2021*).

We recently described a unique example of functional visual field partitioning in the hummingbird hawkmoth *Macroglossum stellatarum*. This insect uses translational optic flow only in their ventral and lateral visual field for flight guidance, while the same stimuli in the dorsal visual field evoke a directional response (*Bigge et al., 2021*; *Figure 1A*). In this directional response, the hawkmoths orient their flight along the main axis of different types of patterns, which is reminiscent of the dorsal orientation responses of ants under tree canopies (*Hölldobler, 1980*). Yet, the nature of the dorsal response in hawkmoths remains largely elusive, lacking a comprehensive comparison of its features with the well-described translational optic flow response.

The behavioural partitioning of the visual field matches the prevalence of translational optic flow and contrast cues across the different regions of the visual field (*Bigge et al., 2021*; *Calow and Lappe, 2007*; *Zanker and Zeil, 2005*; *Figure 1B*). Unlike other examples of visual field partitioning, such as landing and predator avoidance to looming stimuli in fruit flies (*Tammero and Dickinson, 2002*), the hawkmoth flight responses to dorsal and ventrolateral visual cues are not functionally exclusive, suggesting these processes might act in parallel. This poses the question whether the integration of the two systems also corresponds with the natural cue statistics when both systems are active at the same time.

To address these questions, we characterised the unique directional response of the hummingbird hawkmoth to compare it to translational optic flow-based flight responses using behavioural investigations in flight tunnels. To dissect the integration of the dorsal response and ventrolateral optic

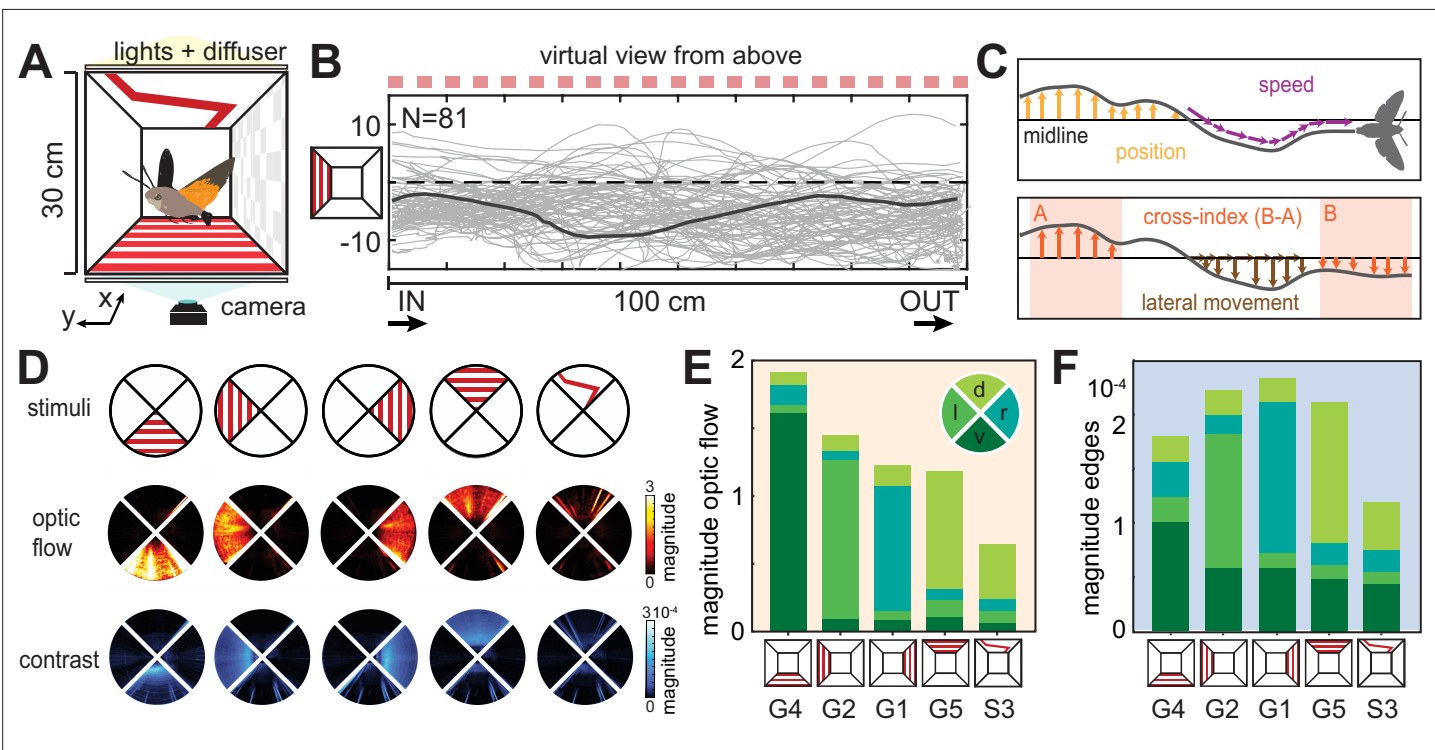

**Figure 2.** Optic flow-based flight control and dorsal directional responses. (**A**) Hawkmoth flight behaviour was tested in a 100 cm long, 30-by-30 cm wide and high tunnel, in which visual patterns could be presented on any side to generate translational optic flow and directional cues. (**B**) The hawkmoths' flight paths (virtually rotated to be shown from above the tunnel, with a grating pattern on the left tunnel side indicated as red squares) were digitised with a camera mounted below the tunnel. N gives the number of flights. The dark grey line highlights one representative path. (**C**) From the flight paths, we quantified the median position off the midline, the average frame-by-frame speed, the proportion of lateral movement (as the ratio of frame-by-frame movement perpendicular and parallel to the longitudinal axis of the tunnel), and the cross-index; the difference in lateral position in the first and last third of the tunnel. (**D**) Heatmaps of the magnitude of translational optic flow (middle row) and contrast edges (bottom row) in the different tunnel conditions (top row) used in conflict experiments. (**E**, **F**) Stacked bars present the average magnitude of translational optic flow (**E**) and contrast edges (**F**) in each of the four quadrants (ventral, left, right, and dorsal) in the five tunnel conditions depicted in (**D**). For the letter-number identifiers of each condition, see *Supplementary file 1*.

flow-based flight control, we applied cue conflict experiments, as are commonly used in multisensory integration studies (*Dacke et al., 2019*; *Stöckl et al., 2016*). We set these results into context with natural visual scene statistics to understand the ecological relevance of the integration hierarchies of the parallel pathways for flight guidance in the hummingbird hawkmoth.

## Results

We monitored the hawkmoths' behaviour in flight tunnels, which they traversed on their way between two cages with hidden feeders (*Figure 2A*). To generate visual stimulation, we used semi-transparent red plastic sheets, through which the animals could be filmed and tracked (see 'Materials and methods', for an overview of all conditions and condition identifiers, *Supplementary file 1*). From the tracked and digitised flight paths (*Figure 2B*), we extracted features of their flight responses: the average frame-by-frame flight speed, the position relative to the midline, the relative proportion of movement along the long versus short axis of the tunnel, and the cross-index (difference in lateral position at the exit versus entrance, *Figure 2C*). We measured the magnitude of translational optic flow and contrast cues generated by the visual stimuli we used to evaluate the behavioural responses of the animals with respect to the external stimulus strength (*Figure 2D–F*). Due to the transparency of the tunnel floor and ceiling, and the lighting from above, the magnitude of translational optic flow (*Figure 2E*) and contrast cues (*Figure 2F*) was not exactly the same for the four sides of the tunnel, despite using the same set of stimuli.

### Response characteristics to dorsal versus ventrolateral stimuli in isolation

When gratings were presented on a lateral tunnel wall, oriented perpendicular to the tunnel's major axis to generate translational optic flow (henceforth called optic flow) by the animals' own movement through the tunnel, the hawkmoths kept a perpendicular distance to the respective walls, resulting in significant shifts in the median position relative to the midline when only one wall generated optic flow (*Figure 3A*, G1 and G2), or a centring of flight paths around the midline when both walls generated optic flow (*Figure 3A*, G3). We indirectly measured the relative vertical position of the hawkmoths in the tunnels, and thereby their distance to dorsal and ventral patterns, by the size of the moths' image in the videos (see 'Materials and methods'). While this indirect measure must be treated with caution, the general trends are indicative of optic flow responses: hawkmoths flew higher, and thus further away from ventral perpendicular gratings compared to flights without patterns (*Figure 3—figure supplement 1D*, G4 versus N). This was also the case when the perpendicular gratings only extended over one side of the ventral tunnel (*Figure 3—figure supplement 1H*, H1 and H2). Moreover, hawkmoths flew at a similar height with longitudinal ventral gratings (which generated little optic flow when the moths flew along the tunnel's major axis, *Figure 5—figure supplement 1C*, L1) as without patterns (*Figure 5—figure supplement 1C*, N), and significantly lower than with perpendicular ventral gratings (*Figure 5—figure supplement 1C*, G10). Thus, both for lateral and ventral cues, hawkmoths kept a greater perpendicular distance from high compared to low optic flow patterns.

Previous work has shown that the distance hawkmoths keep to lateral optic flow depends on its spatial frequency and contrast, and is the largest for the optimal frequency and highest contrast, while the effect of lateral optic flow on flight speed and flight straightness remained similar for all patterns that hawkmoths could perceive (*Bigge et al., 2021*; *Stöckl et al., 2019*). With the limited range of spatial frequencies tested for ventral patterns, we did not observe a significant difference in perpendicular distance for different spatial frequencies (*Figure 3—figure supplement 1G*). This might have been due to differences in the pattern structure (previous studies used sinewave and checkerboards rather than square-wave gratings), or the very coarse measure of distance applied here. Like lateral optic flow patterns (*Figure 3B and C*, G1–G3), ventral optic flow patterns induced a reduction in flight speed (*Figure 3B*, *Figure 3—figure supplement 1F*) and lateral movement (*Figure 3C*, *Figure 3—figure supplement 1C*). The magnitude of lateral movement varied significantly with the spatial frequency of the ventral patterns (*Figure 3—figure supplement 1C*, G4, G8–G9), and the speed with the finest spatial gratings was numerically larger, though not significantly different (*Figure 3—figure supplement 1C*, G8 versus G4, G9), suggesting that like for lateral patterns, the optic flow-based response to ventral stimuli could be spatial frequency dependent.

**Figure 3.** Optic flow-based flight control and dorsal directional responses. (**A–C**) Median position, average speed, and proportion of lateral movement with grating patterns on either tunnel side. (**D**) Cross-index with a red stripe which changed its position in the central third of the tunnel, crossing from one tunnel side to the other. The last two conditions present a version of this stripe, which repeated at the same spatial frequency as the grating patterns. (**E**) Proportion of lateral movement with gratings of different spatial frequencies (repeating every 3 cm, 6 cm, and 12 cm), mounted to the tunnel ceiling. (**F**) Median position of flight tracks with gratings perpendicular (generating strong translational optic flow) and parallel (weak translational optic flow) to the tunnel's longitudinal axis, covering one side of either the tunnel ceiling or floor. Black letters show statistically significant differences in group means or median, depending on the normality of the test residuals (see 'Materials and methods', confidence level: 5%). The red letters in (**A**) represent statistically significant differences in group variance from pairwise Brown–Forsythe tests (significance level 5%). Conditions with different letters were significantly different from each other. The white boxplots depict the median and 25–75% range, the whiskers represent the data exceeding the box by more than 1.5 interquartile ranges, and the violin plots indicate the distribution of the individual data points shown in black. For the letter-number identifiers below each condition, as well as each sample size, see *Supplementary file 1*.

The online version of this article includes the following figure supplement(s) for figure 3:

**Figure supplement 1.** Optic flow-based flight control and dorsal directional responses.

Directional responses were characterised by a much stronger reduction in flight speed than observed for lateral or ventral optic flow cues (*Figure 3B*, G5) and highly increased lateral movement (*Figure 3C*, G4). This increased lateral movement was caused by an alignment of the hawkmoths' flight direction with the main axis of the dorsal cue. It was most clearly observable with a stimulus that crossed from one side of the tunnel to the other, which resulted in an according change in the hawk-moths' lateral position across the tunnel (*Figure 3D*, S3 and S4). This directional flight behaviour was never observed with ventral switch stimuli (*Figure 3D*, *Figure 3—figure supplement 1*, *Figure 4—figure supplement 1*). To rule out that the dorsal directional behaviour was an avoidance response

of the dorsal coverage generated by the stripe and was indeed a true alignment with the stripe's direction, we presented a grating of switching lines, which uniformly covered the entire tunnel ceiling. It elicited qualitatively the same switching response as single stripes (*Figure 3D*, S5 and S6). More-over, with dorsal gratings of different spatial frequencies, hawkmoths showed a proportional change in lateral movement, consistent with following the stripes' directional information, more so for higher spatial frequencies (*Figure 3E*, G5–G7). For ventral gratings, the proportion of lateral movement and flight speed did not show a consistent variation with stripe frequency (*Figure 3—figure supplement 1C*, G4, G8–G9).

Together with increasing lateral movement, hawkmoths reduced their flight speed moderately even for a single dorsal line stimulus, which covered only a small portion of the dorsal visual field (*Figure 3—figure supplement 1A*, S1–S2) and drastically for all dorsal contrast patterns that covered at least half the tunnel ceiling, even if they did not generate strong optic flow cues (*Figure 3—figure supplement 1A , B and I* , S3–S6, H3–H8, G5–G7). If possible, hawkmoths strictly avoided flying under dorsal coverage of the tunnel (*Figure 3F,* H3–H8). They avoided cues that contrasted strongly against the flight direction more than those with little contrast changes (*Figure 3F*, H4 and H7 versus H3 and H6). This also became evident when directly comparing gratings aligned perpendicular and parallel to the flight direction as hawkmoths changed their median flight position in the tunnel to avoid the gratings perpendicular to the flight direction (*Figure 3F*, H5 and H8). On the other hand, hawkmoths did neither avoid nor seek out flying over contrasting structures in the ventral visual field

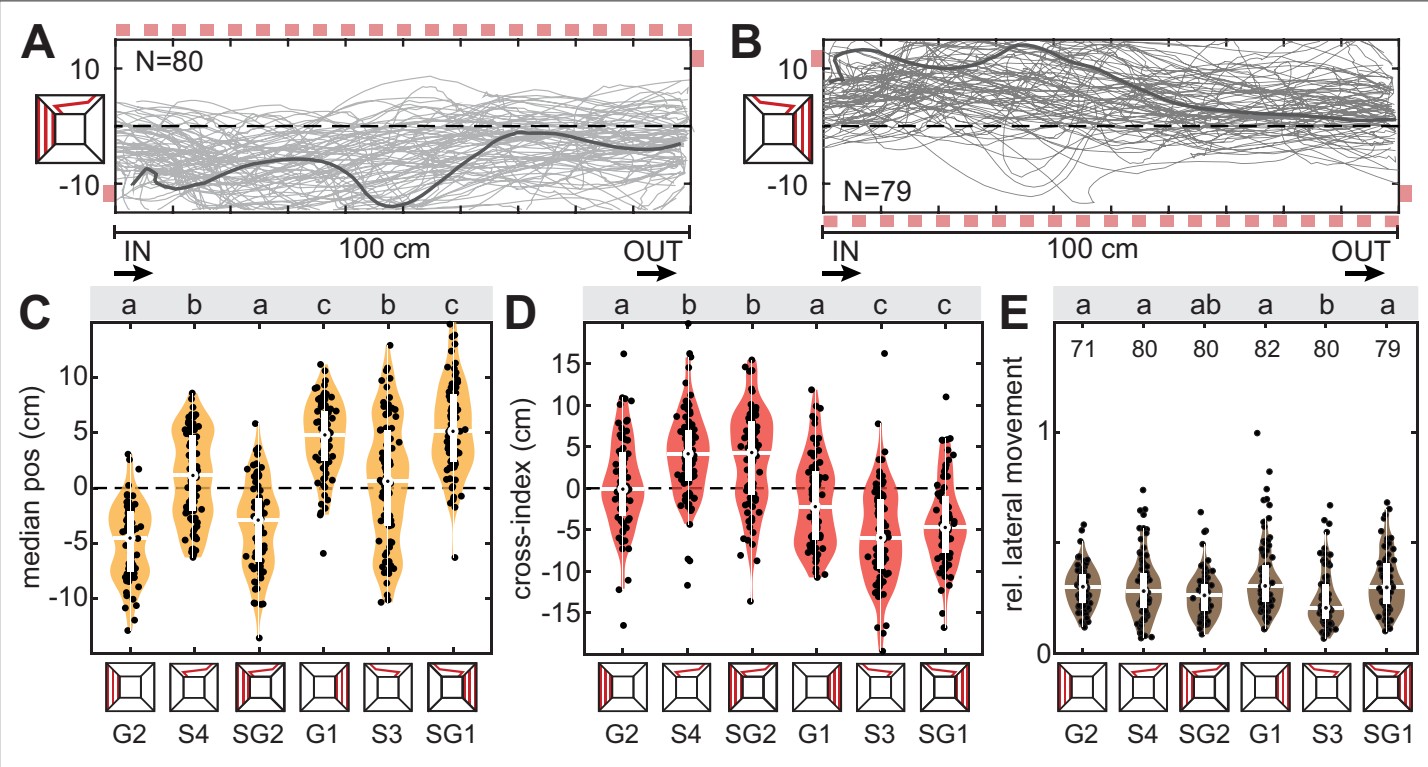

**Figure 4.** Cue conflict: lateral optic flow and dorsal directional cues. (**A, B**) Flight tracks of hawkmoths virtually rotated to be shown from above the tunnel, which were presented with a lateral grating inducing translational optic flow indicated on the top in (**A**) and bottom in (**B**), and a dorsal line which switched sides from the first to the last third of the tunnel (the start and end points are indicated with red squares). N represents the number of flights. The dark grey line highlights a single representative flight track. (**C–E**) Median position, cross-index, and proportion of lateral movement with either grating patterns, the dorsal line stimulus, or a combination of both. Black letters in (**C–E**) show statistically significant differences in group means or median, depending on the normality of the test residuals (see 'Materials and methods', confidence level: 5%). Conditions with different letters were significantly different from each other. The white boxplots depict the median and 25–75% range, the whiskers represent the data exceeding the box by more than 1.5 interquartile ranges, and the violin plots indicate the distribution of the individual data points shown in black. For the letter-number identifiers below each condition, as well as each sample size, see *Supplementary file 1*.

The online version of this article includes the following figure supplement(s) for figure 4:

**Figure supplement 1.** Cue conflict: lateral optic flow and dorsal directional cues.

(*Figure 3F* , H1–H2). Interestingly, unlike for lateral and ventral gratings, hawkmoths did not keep a greater perpendicular distance to dorsal gratings (*Figure 3—figure supplement 1D and G* , S3C, G5–G7, H3–H8). On the contrary, they flew closer to the tunnel ceiling in these conditions than in tunnels without patterns, and mostly at a similar height than with ventral patterns. In tunnels with dorsal gratings covering one side, in which the moths flew exclusively on the free side, they remained at a similar height than in tunnels without patterns (*Figure 3—figure supplement 1I*, , N versus H4 and H7).

Taken together, these observations suggest that the hummingbird hawkmoths' directional responses to contrast patterns in the dorsal visual field did not bear hallmarks of optic flow-based flight response, as in the ventrolateral visual field. This strongly suggests that a different visuomotor pathway underlies this flight behaviour. We therefore asked next, whether the responses to dorsal versus ventrolateral stimuli resulted from independent control systems, which act in parallel.

## Cue conflict: Lateral optic flow and dorsal directional cues

To test whether and how optic flow-based avoidance and dorsal direction-following combine to guide the flight position of hawkmoths, we generated a direct conflict by combining a dorsal directional and lateral optic flow cue, which both induce hawkmoths to fly to the same side of the tunnel (*Figure 4A and B*). For the optic flow cue, we presented vertical red stripes on one tunnel side, which the animals increased their distance to, when presented on their own (*Figure 4C*, G1 and G2). The dorsal directional cue was a red stripe, which switched tunnel sides towards the optic flow cue in the travel direction of the animals. On its own, this cue induced a cross-over in median tunnel position (*Figures 4D*, S1 and S2). When both cues were presented in combination, we observed mixed responses: the hawkmoths kept their distance to the optic-flow tunnel side with the same magnitude as before (*Figure 4D*, SG1 and SG2). Simultaneously, in the now more limited lateral space of half the tunnel width, they performed the switching behaviour, resulting in as high a cross-index as with only the dorsal directional cue alone (*Figure 4E*, SG1 and SG2). This is remarkable because it shows the animals were not simply following the red stripe as they were not flying under it after the switch to the other tunnel side occurred, but they followed the directional information of this stimulus. Thus, hawkmoths performed both optic flow-based distance regulation and dorsal directional manoeuvres at the same time, suggesting that these two control systems act in parallel, with similar weights on the final response. Moreover, these weights did not reflect the magnitude of translational optic flow or contrast edges generated by the two types of cues. The lateral grating presented a much higher amplitude of translational optic flow, as well as a higher magnitude of contrast averaged along the length of the tunnel (*Figure 2E and F*).

To test whether the parallel action of the optic flow and dorsal directional control systems generalised across tasks and visual stimulation types, we tested hawkmoths with a variant of the optic flow-based lateral distance regulation, as well as a different contrast pattern: we used a centring task, as frequently used in other species (*Collett, 2002*; *Egelhaaf, 2023*; *Srinivasan et al., 1999*), by presenting black-and-white checkerboard patterns on both lateral tunnel walls. These were combined with the same dorsal red stripe that changed its lateral position between the first and last third of the tunnel (*Figure 4—figure supplement 1*). The checkerboard pattern on its own induced a centring response (*Figure 4—figure supplement 1A*, C1) and straight flights with a very low cross-index (*Figure 4—figure supplement 1B*, C1). The dorsal direction-switching red stripe presented on its own resulted in a significantly wider positional distribution across the tunnel width (*Figure 4—figure supplement 1A*, S3 and S4), as well as a high cross-index (*Figure 4—figure supplement 1B*, S3 and S4). Presenting both cues combined resulted in a response combination of both optic flow and directional control: the animals centred as strongly as in the optic flow-only condition (*Figure 4—figure supplement 1A*, SC1 and SC2), and their cross-index was similar in direction to the dorsal directional-only condition (*Figure 4—figure supplement 1B*, SC1 and SC2). The mixed response also manifested in flight speed, which was significantly lower for dorsal directional-only cues than optic flow-only cues and had an intermediate value for the mixed conditions (*Figure 4—figure supplement 1C*, SC1 and SC2).

Thus, hawkmoths performed optic flow-based flight control, including lateral distance regulation and speed control in parallel with dorsal directional manoeuvres across different visual stimuli.

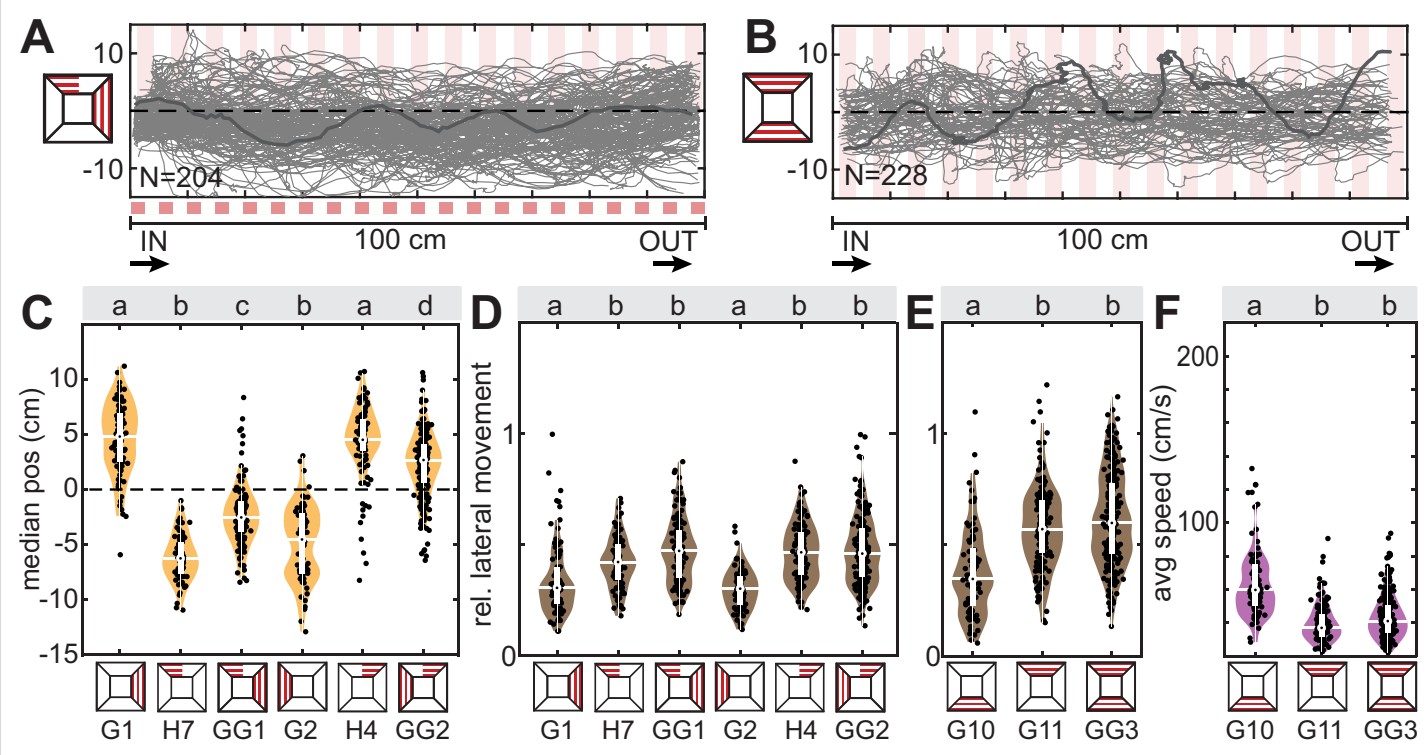

**Figure 5.** Cue conflict: lateral versus dorsal avoidance. Flight tracks of hawkmoths virtually rotated to be shown from above the tunnel, which were presented with (**A**) a lateral (red squared) and a dorsal grating pattern spanning one half of the tunnel (semi-transparent red stripes), and (**B**) a ventral and dorsal grating pattern. N represents the number of flights. The dark grey line highlights a single representative flight track. (**C–F**) Median position, proportion of lateral movement and flight speed with either pattern in isolation and in combination. Black letters in (**C–E**) show statistically significant differences in group means or median, depending on the normality of the test residuals (see 'Materials and methods', confidence level: 5%). Conditions with different letters were significantly different from each other. The white boxplots depict the median and 25–75% range, the whiskers represent the data exceeding the box by more than 1.5 interquartile ranges. The violin plots indicate the distribution of the individual data shown in black. For the letter-number identifiers below each condition, as well as each sample size, see *Supplementary file 1*.

The online version of this article includes the following figure supplement(s) for figure 5:

**Figure supplement 1.** Cue conflict: lateral distance regulation versus dorsal avoidance.

## Cue conflict: Lateral versus dorsal avoidance

In the previous set of conflict tests, both lateral optic flow-based distance regulation and dorsal directional responses could be enacted with the same strength as when presented individually. We next tested how the animals responded when dorsal and lateral responses were conflicted, so that it was not possible to enact both to their original magnitude simultaneously. To test this, we set the lateral distancing and dorsal avoidance responses in conflict by presenting a grating of vertical red stripes on one tunnel wall, which induced the animals to fly on the opposite side of the tunnel, coupled with a grating of the same type on one half of the tunnel's ceiling (*Figure 5A and B*). On its own, the animals avoided the dorsal tunnel coverage by flying on the uncovered tunnel side (*Figure 5C*, H4 and H7). Similarly, animals avoided the lateral tunnel side that presented a grating in isolation (*Figure 5C*, G1 and G2). When combined, we observed a clear hierarchy in responses: the vast majority of animals flew on the side of the tunnel presenting lateral optic flow, thus avoiding the dorsally covered section (*Figure 5C*, GG1 and GG2). Yet, the animals' median position was also significantly different from the dorsal-only condition and shifted closer towards the lateral-only condition (*Figure 5C*, GG1 and GG2). Moreover, not a single animal had their median flight position in the covered tunnel side in the dorsal-only condition, but a subset of animals did in the conflict situation (*Figure 5C*, GG1 and GG2). Thus, the animals responded to both cues, though with a higher weight for dorsal avoidance. This weighting of responses did not reflect a higher magnitude of translational optic flow or overall contrast in the dorsal versus the lateral visual field (*Figure 2E and F*).

Noticeably, in these conflict experiments, the lateral movement of the animals (*Figure 5D*, GG1 and GG2), and their flight speed (*Figure 5—figure supplement 1A*, GG1 and GG2) was significantly different from the lateral optic flow only condition, but not from the dorsal-only condition. High lateral movement and very low speed were highly indicative of the dorsal directional responses and suggest that even in the presence of lateral optic flow in the conflict situation, which supports straight, moderately fast flight, the animals were strongly guided by dorsal cues. Since we previously found that the animals' flight straightness and speed was guided stronger by ventral than lateral optic flow cues (*Bigge et al., 2021*), we simultaneously presented dorsal and ventral gratings to test whether dorsal cues would also take over flight guidance in combination with ventral cues. In the conflict situation, the lateral movement (*Figure 5E*, GG3) and flight speed (*Figure 5F*, GG3) were significantly different from the ventral-only condition (G10), but not significantly different from the dorsal-only one (G11). This demonstrates that in the presence of dorsal contrast cues, the optic flow-based control on flight straightness and speed was taken over by dorsal directional guidance, suggesting the latter has a greater weight in the control hierarchy. This was observed despite the fact that the ventral gratings generated about 50% stronger translational optic flow than dorsal ones, and a comparable magnitude of contrast cues (*Figure 2E and F*). Yet, when analysing the relative flight height (*Figure 5—figure supplement 1C*), we found that the hawkmoths kept a similar distance to ventral patterns in the conflict situation (GG3) than with ventral cues only (G10), and a significantly higher flight position than with dorsal-only cues (G11), suggesting that similar to lateral optic flow cues, the perpendicular distance regulation to ventral optic flow cues remained intact when dorsal cues were presented simultaneously.

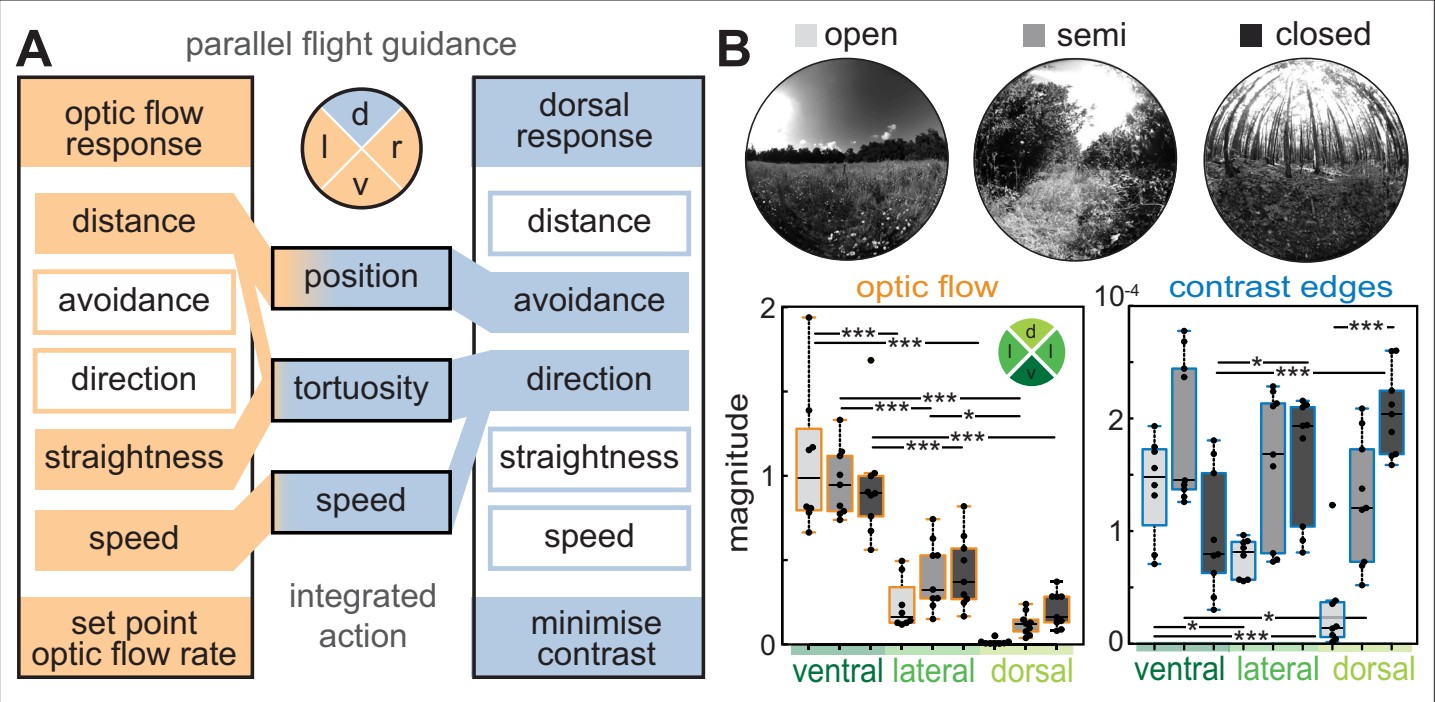

**Figure 6.** Integration of optic flow and directional responses in the context of natural visual cues. (**A**) The response features of translational optic flow-based flight control and the dorsal response in the hummingbird hawkmoth. Filled boxes represent response features we observed for the respective control systems, and empty boxes represent features we did not observe in response to ventrolateral optic flow cues, or dorsal cues, respectively. Which feature within each system contributed to the key flight parameters we measured is depicted by the coloured links between the feature boxes and flight parameters. Both systems acted on flight control in parallel and produced mixed responses of equal magnitude as either system individually. Only when a mixed response of equal strength was not possible, did we observe a hierarchy between the systems. Its effect on the key flight parameters for the stimuli we tested is qualitatively depicted by the gradient of colours. (**B**) The distribution of optic flow and contrast cues across different habitat types: open (no bushes or trees within 500 m of the camera), semi-open (lateral vegetation but no closed canopy), and closed (entirely closed canopy), with one example image per habitat type. Boxplots depict the mean magnitude of translational optic flow (left panel) and contrast edges (right panel) across habitat types, and for three different scenes within habitats in the dorsal, ventral, and lateral segments of the visual field (see coloured inset). The boxplots depict the median and 25–75% range, the whiskers represent the data exceeding the box by more than 1.5 interquartile ranges, and the individual data points are shown in black. Statistical results from a linear mixed-effects model (see Methods) are abbreviated as *p<0.05, **p<0.01, ***p<0.001.

## Discussion

In this study, we characterised the dorsal directional flight response of hummingbird hawkmoths and revealed that it is fundamentally different from the 'canonical' optic flow-based flight control (*Srinivasan et al., 1999*). Using conflict experiments, we demonstrated that optic flow-based and dorsal directional flight responses act in parallel, though with different relative weights.

### The nature of the dorsal response

Our results demonstrate that the flight responses of hawkmoths to dorsal stimuli did not bear the typical hallmarks of translational optic flow-based flight control (*Figure 6A*) as it is known from bees (*Baird et al., 2005*; *Baird et al., 2010*; *Dyhr and Higgins, 2010*; *Kirchner and Srinivasan, 1989*), flies (*David, 1982*; *Fry et al., 2009*; *Portelli et al., 2011*; *Srinivasan et al., 1999*), or moths (*Vickers and Baker, 1994*; *Willis and Card, 1990*; *Kennedy and Marsh, 1974*; *Kuenen and Baker, 1982*; *Stöckl et al., 2019*) to ventrolateral stimuli, and for honeybees to dorsal ones as well (*Portelli et al., 2011*). In this paradigm, the magnitude of translational optic flow is thought to be kept constant by a reduction of speed and an increase in perpendicular distance to cues generating optic flow (*Portelli et al., 2011*; *Franceschini et al., 2007*). While the hawkmoths in our experiments slowed down upon perceiving translational optic flow dorsally, they did so much more than for the same stimuli ventrolaterally. Moreover, they aligned with the dominant contrast axis of the stimuli and thus generated a much higher proportion of lateral movement for gratings perpendicular to the tunnel axis. As previously shown in these hawkmoths, high degrees of lateral movement correlate with low flight speed (*Stöckl et al., 2019*), so that the low speed was likely a side effect of the lateral flight, not a response to the optic flow. This also fits the observation that the hawkmoths flew distinctly faster with longitudinal dorsal stripes (*Figure 5—figure supplement 1B*, L1).

Given the hawkmoths' flight responses, one might hypothesise that contrast in the dorsal visual field induced a behaviour aimed at minimising optic flow, rather than keeping it constant, as they would ventrolaterally. What fits the hypothesis is that animals avoided flying under any visual contrast when possible. If this was not possible, they flew under the less-optic flow-inducing portions of the tunnel (*Figure 3F*, H5 and H8). Aligning with the dominant contrast axis could be a strategy to minimise the perceived translational optic flow. However, hawkmoths did not show a perpendicular distance regulation in response to dorsal optic flow cues as they did for ventrolateral ones. If anything, they flew closer to dorsal stimuli than in a tunnel without patterns (*Figure 3—figure supplement 1D*, N versus G5). This suggests that the dorsal response is not an optic flow reduction response, but rather aims at minimising the perceived contrast (*Figure 6A*).

We therefore hypothesise that the dorsal response might be an adaptation to avoid flying under canopies: both the strong avoidance of any dorsal coverage, whether inducing translational optic flow or not, and the alignment with dominant contrast edges, which could be canopy borders, could help to remain flying in the open. The lack of a distance regulation to dorsal optic flow cues might also be explained by a canopy avoidance strategy: if hawkmoths avoided flying under canopies, they would rarely encounter any dorsal optic flow. All the more so, since the natural habitat of these hawkmoths is described as mostly open and semi-open (*Pittaway, 1993*). Moreover, even if a hawkmoth was trapped under dorsal coverage, their most common escape strategy is flying upwards towards a bright spot of sky, which would be impaired if the moths had a regulatory mechanism like for ventrolateral optic flow, which kept them at a safe distance from optic flow-inducing structures.

### Integration of the dorsal system with optic flow-based flight control

For ventrolateral flight responses of hummingbird hawkmoths, we confirmed previous studies, which show typical translational optic-flow-based flight regulation (*Grittner et al., 2022*; *Stöckl et al., 2019*; *Bigge et al., 2021*; *Figure 6A*). Our vertical distance measure added a final puzzle piece to this and confirmed that as for lateral cues, hummingbird hawkmoths also increased their perpendicular distance to ventral translational optic flow (*Figure 3—figure supplement 1D*, G1 and S1G, G4, G8–G9, and S1H, H1–H2, versus N).

Our results suggest that this optic flow-based flight control system acts in parallel to the dorsal canopy avoidance system because combined stimuli in both the dorsal and ventral visual field created mixed responses, which are only possible if both systems were active at the same time. The animals enacted responses to both systems to the same degree that they occurred for each system in isolation,

if the space and stimulus configuration permitted (*Figure 3*, *Figure 4—figure supplement 1*). This was robust across different stimulus geometries (translational optic flow cues presented on one or both walls), as well as stimulus features (red single frequency gratings and black and white random checkerboards that contained a range of spatial frequencies).

A response weighting was only observable in situations in which responses to both systems were not possible at original strength (*Figure 6A*). Our series of conflict experiments revealed a response hierarchy between the two systems, which varied for different features of the flight responses even for the same stimulus combination: in terms of flight position, the highest weight was given to the avoidance of dorsal coverage. This behaviour occurred simultaneously with the optic flow-based distance regulation, but reduced the latter's impact when set in direct conflict. Importantly, the magnitude of dorsal contrast avoidance was also significantly reduced when in conflict with lateral avoidance, yet not to the same degree as lateral distance regulation (*Figure 4*, H7 and H4 versus GG1 and GG2 versus G1 and G2). Moreover, when dorsal avoidance was not possible, the perpendicular distance regulation to ventral cues operated to the same degree as for isolated stimuli (*Figure 5—figure supplement 1C*, G10 versus GG3). Flight speed and path straightness were dominated by the response to dorsal cues as they had the same magnitude as the dorsal cues in isolation for all stimulus combination with dorsal grating stimuli.

Since the response hierarchy between the systems only occurred when responses to both pathways could not be enacted to their original magnitudes, we would predict that the integration weights vary with the geometry of the moths' environment and the distribution of the relevant cues, as these will determine how strongly the animals respond to either the dorsal or ventrolateral system.

Our results also question whether hawkmoths use a combination of flight speed and distance regulation to retain a constant optic flow percept, as has been suggested in other insects (*Portelli et al., 2011*; *Franceschini et al., 2007*; *Serres and Ruffier, 2017*). Indeed, the hawkmoths in our experiments kept a similar distance regulation with ventral and lateral optic flow, despite drastically different flight speeds (which were induced by the presence or absence of dorsal cues, *Figure 5F*, *Figure 5—figure supplement 1C*). Instead, these results suggest that the animals can measure the distance to optic flow-generating textures in their ventrolateral visual field and subsequently adjust their distance, independently of speed. This could be possible using optic flow divergence, the perceived expansion and contraction caused by flight movements towards and away from a texture (*Bergantin et al., 2021*; *Ho et al., 2017*). If hawkmoths measured the distance to objects in their ventrolateral visual field using expansion cues and separately attempted to retain the perceived optic flow constant via adjustments of flight speed, this would explain the two modules having different weights in the control hierarchy and operating separately from each other in the behavioural experiments.

## The ecological relevance of the dorsal and ventrolateral flight control systems

If one interprets the observed flight control hierarchy in terms of ecological function, the highest priority for hummingbird hawkmoths would be to avoid flying into foliage. To this aim, they avoid any dorsal coverage. This aim operates in parallel with keeping a distance to ventrolateral texture, to remain at a safe distance from potential obstacles. Interestingly, the tolerated distance to ventrolateral texture was reduced at the benefit of avoiding dorsal coverage. It remains to be tested if there is a minimum tolerated distance that the animals consider to be safe.

The dorsal contrast avoidance behaviour has not yet been observed in other insects. It is possible that the lifestyle of the insects tested previously is more tolerant of surrounding foliage than that of hummingbird hawkmoths: honeybees naturally build their nests in trees, while bumblebee nests in the ground might be surrounded by trees or shrubs. The animals thus encounter, and might also use, surrounding foliage, including dorsal coverage, during their daily navigation to and from the nest (*Collett and Baron, 1994*; *Ibarra de et al., 2009*). They might therefore be tolerant of dorsal contrast cues; and because they encounter them frequently, use their translational optic flow component to guide flight in the same way across the entire visual field. To understand whether using or avoiding dorsal optic flow is indeed coupled with insect habitat and lifestyle, comparative studies are required, particularly of hawkmoth species that live in forest versus open environments (*Pittaway, 1993*), and of other insects that live in similar open habitats to hummingbird hawkmoths.

One other explanation for the observation that honeybees keep the perceived translational optic flow constant in either part of their visual field by adjusting distance and speed (*Portelli et al., 2011*) is odometry. Honeybees have been shown to measure the distance of their path to food sources using the translational optic flow integrated during their flight and communicate this information to nestmates to guide them to the same food source (*Srinivasan et al., 1997*; *Srinivasan et al., 2000*). In order for this strategy to work, reproducible optic flow on consecutive flights is required – which is not straightforward, since different landscape structures, and flight at different speeds and distances from the surrounding structure, generate different perceived magnitudes of translational optic flow (*Tautz et al., 2004*; *Figure 6B*), resulting in different perceived lengths of path travelled. One way to generate reliable distance measures is to keep perceived translational optic flow constant, which might be a key function for the translational optic flow regulator observed in bees. This hypothesis fits well with the observation that honeybees, when trained to fly close to the one side of a tunnel on the way to a food source (*Serres et al., 2008*), will repeat this strategy when the entrance into the tunnel, or the tunnel shape, changes (*Portelli et al., 2017*), thus preserving the rate of optic flow they perceived when learning their way to the food source. Thus, in honeybees, keeping the perceived translational optic flow constant across the entire visual field might have less to do with flight guidance or safety, and more with keeping a reliable translational optic flow percept for odometry. This would also explain the differences to hawkmoths, which are not known to path integrate, and demonstrate that retaining safe flight is possible without a constant translational optic flow strategy. Comparing path integrating and non-path integrating insects would reveal whether the optic flow regulator strategy involving speed and distance regulation serves path integration rather than flight control.

## General takeaways for parallel control in sensory systems

Beyond the mechanisms of flight control in insects, this study also provides fundamental insights into strategies of partitioning the visual field, and how parallel visuomotor pathways are integrated for a robust behavioural response. The benefit of our study system is that the same cues activate different control pathways in different regions of the visual field, so that the resulting behaviour can directly be interpreted in terms of integration weights (unlike for different qualitative types of visual cues, or different sensory modalities, where comparing the perceived magnitude of the different cues is not straightforward). What these experiments show is that the partitioning of the visual field followed the probability with which these cues occur in the animals' habitat: translational optic flow occurred more frequently and with a higher magnitude in the ventrolateral visual field than in the dorsal one (*Figure 6B*). This was mirrored in the receptive areas of the optic flow-based flight guidance, while canopy avoidance responses occurred exclusively to cues in the dorsal visual field – where contrast cues were the strongest in natural environments with canopies due to the backlighting of the sky (*Figure 6B*). It remains to be tested at what elevation the visual field partitions, and whether there is an area of overlap in which either behaviour could be elicited, as is the case for landing and predator avoidance towards looming stimuli in fruit flies (*Tammero and Dickinson, 2002*). Moreover, visual feature distributions should be measured in more diverse natural habitats, and particularly at different elevations, which represent insect flight altitudes (*Degen et al., 2022*).

Contrary to the visual field partitioning by the stimulus statistics in natural habitats (*Baden et al., 2013*; *Hornstein et al., 2000*; *Turner et al., 2019*), the integration hierarchy did not match the prevalence of natural cues. In the natural habitats of hawkmoths, dorsal coverage was much less frequent than ventrolateral structures generating translational optic flow, yet the hawkmoths responded with a higher weight to the former. Moreover, in our flight tunnel experiments, the animals responded with the same or higher weights to dorsal cues, which had a lower magnitude of translational optic flow and contrast than the same cues in the ventrolateral visual field (*Figure 2E and F*). This differs from other sensory integration strategies, which are generally thought to weigh the modality highest which occurs most frequently, and thus has the highest reliability, or the one that provides most information (*Dacke et al., 2019*; *Stöckl et al., 2016*). The strategy emerging for flight control in the hummingbird hawkmoth is one of reacting to the most immediate 'danger' to the animal: the stimuli that carry the biggest potential threat for the animals' (flight) safety are weighed highest, while in the event of these stimuli, the tolerance for other risks (such as flying closely to lateral structures) is increased. Thus, dorsal contrast cues might not constitute the most frequent input, but the one with the highest

priority in terms of the animals' safety, and therefore are assigned the highest weights in the control hierarchy.

Thus, to understanding parallel processing strategies, and the integration of parallel information for behavioural guidance, both the prevalence and magnitude of the input statistics are important, as well as the relevance of these stimuli to the particular behaviour of the animals. In the case of flight control in the hummingbird hawkmoth, prevalence shapes the partitioning of the visual field, while relevance determines the integration hierarchy of the parallel pathways.

## Materials and methods

### Optic flow imaging

#### Imaging of natural scenes

To film a 180° frontal visual field panorama, we used the imaging setup designed and described in detail in *Bigge et al., 2021*. In short, we captured translational optical flow in videos using a full-HD digital camera (USBFHD01M, ELP), equipped with a 185° fisheye lens (1.08 mm focal length, BL-5MP010820MP13IR, Vision Dimension). The camera was attached to the front of a 19 V-powered linear actuator with 70 cm extension (Bewinner) and a movement speed of 12 mm/s. This system was mounted onto a tripod ca. 1 m above ground and captured the visual scenery during 70 cm of horizontal forward movement at a constant distance to the ground. Videos were captured with the open-source software ContaCam 7.9.0 software (Contaware) at a rate of 30 frames per second. To correct undesired rotation in the videos, generated by the linear actuator's movement, we tracked three horizontally aligned landmarks in outdoor recordings, which were attached to a second tripod and positioned 3 m in front of the camera, to take up only a small portion of the visual field. The landmark positions were used to correct image rotations in MATLAB 2020a (The MathWorks). Any remaining instability in video position was removed with the video stabilisation filter Deshaker 3.1 (Gunnar Thalin; guthspot.org) in VirtualDub v1.10.4 (virtualdub.org).

Videos were collected on sunny days (individual clouds were tolerated) in three different types of natural habitats: open, semi-open and closed. Open habitats had no bushes or trees within 20 m of the camera position; semi-open environments had different combinations of shrubs and trees but no complete cover overhead and the closed habitats were filmed under closed tree canopies (*Figure 6B*). We collected videos in three different locations of each habitat type, and within each of the locations, we filmed at three different perspectives to not bias the analysis with a specific view (with the exception of one location in the open habitat type, where only two perspectives were filmed, resulting in N=8 for open habitats, and N=9 for semi-open and closed ones).

We used the same setup to measure optic flow and contrast edges in the flight tunnel setup to quantify the magnitudes of the stimuli in situ. This was important since the geometry of the lighting and use of semi-transparent materials for lighting and filming on the dorsal and ventral tunnel walls did not provide a perfectly homogeneous stimulation. Moreover, using the same camera and analysis procedures provided comparable optic flow and contrast measures to the outdoor environments. To do so, the camera was positioned centrally with equal distance to all walls and extended from one tunnel entrance 70 cm into the flight tunnel.

#### Analysis of optic flow fields and contrast features

The dense optical flow in the videos was estimated by the Gunnar Farnebäck method (*Farnebäck, 2003*), using Open CV with Python 3.7.6, which computed an optic flow vector (angle and magnitude) for every pixel in each frame. In Matlab 2022a, we then used a filter to pass only the optic flow vectors that corresponded to translational optic flow along the camera's trajectory. In our setup, the viewing axis and the direction of camera travel were the same, thus translational optic flow was represented in vectors with angles radially oriented around the image centre. The mean magnitude of these vectors was then calculated for each pixel of the video. All pixel magnitudes of each video of a scene were averaged and extracted in one of four quadrants (dorsal, ventral, left lateral, right lateral) for subsequent analysis.

We extracted contrast edges using the MATLAB 2022a inbuilt function 'edge', using a Prewitt filter, and variable contrast threshold (0.01, 0.025, 0.05, 0.1 and 0.5). Each extracted edge in each frame of each video was scaled by the highest threshold required to detect it. All contrast edges of each video

of a scene were averaged and extracted in one of four quadrants (dorsal, ventral, left lateral, right lateral) for subsequent analysis. For natural visual scenes, the lateral quadrants were averaged and combined, while they were left separate for the analysis of optic flow and contrast in the tunnels, to allow for a detailed assessment of the employed tunnel conditions.

Statistical analysis of the magnitude of optic flow and contrast edges across habitat types (open, semi, closed) and quadrants of the visual field (ventral, lateral, dorsal) was performed in R 4.3.1 using a linear mixed-effects model (lm4 package) with scenes within conditions as random effects, using the formula

$$\text{magnitude} \sim \text{habitat} \times \text{quadrant} + (1 \mid \text{scene})$$

We confirmed that the full model had a lower AIC and deviance than the null model with only random effects, as well as models with either factor only and random effects, before continuing pairwise post hoc tests (emmeans package, default correction: Tukey).

## Behavioural experiments
### Animals
Adult male and female *Macroglossum stellatarum* (Linnaeus 1758) were raised on their native host plant *Gallium* sp. The eclosed adults (both male and female) were allowed to fly and feed from artificial feeders in flight cages (60 × 60 × 60 cm, length × width × height) on a 14 h:10 h light:dark cycle for at least one day before experiments began.

### Flight tunnel setup
Details of the experimental setup and data analysis have been described previously (*Bigge et al., 2021*; *Stöckl et al., 2019*) and will be briefly summarised. Two flight cages (60 × 60 × 60 cm, length × width × height) were connected by a Perspex flight tunnel (100 × 30 × 30 cm, length × width × height, *Figure 2A and B*). The tunnel and cages were illuminated by fluorescent tubes with a daylight-like spectrum (Osram L 18W/965 Biolux Tageslicht G13). A white screen in the middle of the left flight cage (60 cm high, 45 cm wide) obstructed the view into the cage when animals were flying in the tunnel. The feeders were hidden behind this screen. A camera (PlayStation Eye, PS3, Sony) was positioned 1.5 m below the tunnel to film its entire length. It was controlled by ContaCam 7.9.0 beta7 software (Contaware) in motion detection mode at a rate of 50 frames per second and an aspect ratio of 640 × 480 pixels.

### Visual stimulation
To disguise the visual panorama above the tunnel, white felt was placed on top of the tunnel ceiling as a diffuser. Gauze (Gazin Verbandmull 8-fach, Lohmann & Rauscher) was placed on the tunnel floor to avoid light reflections and access to the visual panorama below the tunnel (*Figure 2A*). It allowed the camera mounted below the tunnel to film through it. The tunnel side walls were lined with 5% nominal contrast checkerboard patterns (13.9 mm side length, equivalent to a viewing angle of 5.3° from the centre of the tunnel) for all experimental conditions. This provided sufficient visual feedback to reduce collisions of the animals with the walls, while at the same time minimising the classical hallmarks of translational optic flow flight responses (*Bigge et al., 2021*). For stimulation, we used transparent red plastic sheets (Neewer, #10088988), through which the animals could be filmed and tracked using the red channel of the RGB videos. All patterns were mounted so that they were presented on top of the gauze, diffuser, and 5% checkerboard patterns. We constructed grating patterns out of 3 cm wide stripes of the red sheet, alternating with no stripes. These either spanned the entire 30 cm of one tunnel side, or only 15 cm of the ceiling or floor. Additionally, we also constructed 6 cm and 1.5 cm wide striped gratings to test the reliance of the dorsal and ventral responses on the stripe frequency. In addition to the grating stimuli, we constructed a directional stimulus out of the red sheets, which was 3 cm wide, and consisted of a 33 cm straight section on either tunnel side mounted 5 cm from the edge of the tunnel at opposite sides, and a connecting diagonal in the central 33 cm of the tunnel, as well as a version of this stimulus with repetitions of the stripe every 3 cm. We anticipate the red stimuli to only weakly activate the green photoreceptors of *M. stellatarum*, and thus appear as a dark feature on top of a brighter background.

To be able to refer to the individual stimulus conditions in the analysis and results description, we gave each stimulus a unique identifier (see *Supplementary file 1*) and provided these identifiers in the respective figures.

## Experimental procedure

Prior to experiments, 30–40 hummingbird hawkmoth individuals were accustomed to the setup for 2 days as described in *Bigge et al., 2021*. Briefly, during training, the tunnel side walls were lined with a black and white chequerboard pattern for optimal visual feedback. On the first day, feeders were presented freely visible in the left flight cage, and the animals were moved between the left and right flight cage repeatedly. The following day, the feeders were hidden behind a white screen, and the animals were regularly moved back to the right cage which had no feeders, to encourage the use of the tunnel.

After familiarisation with the setup, experiments started with feeders in the left flight cage, which were obstructed from view. Animals that were resting in the left cage were moved back to the right cage (without feeders) every 2 h between 9.00 and 17.00. We previously confirmed for this setup that the majority of animals traversed the tunnel within a period of 3 h, suggesting that the recorded flights were generated by a representative proportion of the hawkmoth group in our experiment (*Bigge et al., 2021*). The visual stimuli were changed daily or, if too few flights were recorded in that period, every 2 days. Flights through the tunnel were filmed continuously, although the first hour after pattern changes was not analysed to allow hawkmoths to adjust to the new visual scenario. To exclude side biases, we performed all directional experiments with patterns on either side of the tunnel.

## Video analysis

As detailed previously (*Stöckl et al., 2019*), not all individuals of *M. stellatarum* traverse the tunnel in a directed fashion. Therefore, we only analysed complete flight tracks from the right to the left flight cage that did not contain landing attempts. Moreover, the tunnel had to be free of other hawkmoths. The flight tracks were then digitised using custom-written software in Matlab R2022a (https://github.com/stoeckl-lab/Bigge_et_al_2025 copy archived at *Stöckl, 2025*), which detected the animals based on frame-to-frame image differences. The resulting image differences contained the silhouette of the hawkmoths, which was used to extract the area covered by each moth in each frame, using the MATLAB inbuilt function regionprops. The animals' current position was extracted as the centroid of each detected area. The size of the area was used as a proxy for the animals' height in the tunnel: the larger the area, the closer the animals flew to the camera, which was positioned under the tunnel; thus, the closer they flew towards the bottom of the tunnel. While this measure did not provide exact flight height in the tunnel (it was sensitive to the animal's posture, including roll and pitch, and motion blur introduced by their speed), it could be used to analyse trends in the average height the moths' traversed the tunnel at (*Figure 3—figure supplement 1D, G, H*, *Figure 5—figure supplement 1C*). The digitised tracks were analysed in the central 80 cm of the tunnel to exclude the portions of flight where cues from the flight cages could have been visible. We quantified the median of the lateral positions (along the width of the tunnel) of each flight track, the median of the frame-to-frame speed of each flight track, the proportion of lateral movement (as the median of the ratio of movement along the lateral to the longitudinal axis of the tunnel), and a cross-index (by subtracting the lateral position of the animal before exiting – 600–800 mm – from the one after entering the tunnel – 200–400 mm, see also *Figure 2C*).

## Statistical analysis and presentation

The median lateral position in the tunnel, lateral movement, cross-position index, and tracking area of individual flights were compared across conditions, using ANOVA with Tukey–Kramer-corrected post hoc comparisons, if the normality of all residuals was confirmed. If not, a Kruskal–Wallis test with Tukey–Kramer–corrected post hoc comparisons was performed. The nature of the test is indicated in the statistical results files in the data repository. If not indicated otherwise, statistical results are shown with a significance level of 5%. The variance in the median lateral position (in other words, the spread of the median flight position) was statistically compared between the groups using pairwise Brown–Forsythe tests, as a measure for the strength of centring in the tunnel.

For the graphical data summary, the individual data points were spread around the x-axis position of the group by drawing from a Gaussian distribution, which was scaled to 0.05 x-axis units. The whiskers of the boxplots represent the data range extending more than 1.5 interquartile ranges from the box limits. The violin plots indicate the distribution of the individual data points; their smoothing kernel was set to 0.04 times the range of each group of data.

## Acknowledgements

We acknowledge funding to A.L.S. by the German Research Council (STO 1255 2-1), and to A.L.S. and R.B. by the Young Scholar Fund of the University of Konstanz. We thank James Foster for suggestions on the statistical analysis.

## Additional information

### Funding

| Funder | Grant reference number | Author |
|---|---|---|
| Deutsche Forschungsgemeinschaft | STO 1255 2-1 | Anna Lisa Stöckl |
| University of Konstanz | Young Scholar Fund | Anna Lisa Stöckl |

The funders had no role in study design, data collection and interpretation, or the decision to submit the work for publication.

### Author contributions

Ronja Bigge, Data curation, Formal analysis, Validation, Investigation, Methodology, Writing – original draft, Writing – review and editing; Rebecca Grittner, Formal analysis, Investigation, Writing – review and editing; Anna Lisa Stöckl, Conceptualization, Resources, Data curation, Formal analysis, Supervision, Funding acquisition, Validation, Visualization, Methodology, Writing – original draft, Project administration, Writing – review and editing

### Author ORCIDs

Ronja Bigge http://orcid.org/0009-0005-0403-4800
Anna Lisa Stöckl https://orcid.org/0000-0002-0833-9995

Reviewer #1 (Public review): https://doi.org/10.7554/eLife.104118.4.sa1
Reviewer #2 (Public review): https://doi.org/10.7554/eLife.104118.4.sa2
Reviewer #3 (Public review): https://doi.org/10.7554/eLife.104118.4.sa3
Author response https://doi.org/10.7554/eLife.104118.4.sa4

## Additional files

### Supplementary files

MDAR checklist

Supplementary file 1. Summary of visual stimulation conditions (labels as used in data repository; https://doi.org/10.6084/m9.figshare.26820091) and number of flight tracks per condition.

### Data availability

Source data for the behavioural experiments, as well as natural scenes and tunnel pattern imaging, and all statistical results files, are available from https://doi.org/10.6084/m9.figshare.26820091. Custom-written analysis code is available from (https://github.com/stoeckl-lab/Bigge_et_al_2025 copy archived at *Stöckl, 2025*).

The following dataset was generated:

| Author(s) | Year | Dataset title | Dataset URL | Database and Identifier |
|---|---|---|---|---|
| Stöckl A, Bigge R, Grittner R | 2025 | Integration of parallel pathways for flight control in a hawkmoth reflects prevalence and relevance of natural visual cues | https://doi.org/10.6084/m9.figshare.26820091 | figshare, 10.6084/m9.figshare.26820091 |

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
