## [Editor Report · eLife Assessment]

This **important** study investigates how hummingbird hawkmoths integrate stimuli from across their visual field to guide flight behaviour. Cue conflict experiments provide **solid** evidence for an integration hierarchy within the visual field: hawkmoths prioritise the avoidance of dorsal visual stimuli, potentially to avoid crashing into foliage, while they use ventrolateral optic flow to guide flight control. These findings will be of broad interest to enthusiasts of visual neuroscience and flight behavior.

---

## [Referee Report · Reviewer #1 (Public review)]

Summary:

Recent work has demonstrated that the hummingbird hawkmoth, Macroglossum stellatarum, like many other flying insects, use ventrolateral optic flow cues for flight control. However, unlike other flying insects, the same stimulus presented in the dorsal visual field, elicits a directional response. Bigge et al., use behavioral flight experiments to set these two pathways in conflict in order to understand whether these two pathways (ventrolateral and dorsal) work together to direct flight and if so, how. The authors characterize the visual environment (the amount of contrast and translational optic flow) of the hawkmoth and find that different regions of the visual field are matched to relevant visual cues in their natural environment and that the integration of the two pathways reflects a prioritization for generating behavior that supports hawkmoth safety rather than the prevalence for a particular visual cue that is more prevalent in the environment.

Strengths:

This study creatively utilizes previous findings that the hawkmoth partitions their visual field as a way to examine parallel processing. The behavioral assay is well-established and the authors take the extra steps to characterize the visual ecology of the hawkmoth habitat to draw exciting conclusions about the hierarchy of each pathway as it contributes to flight control.

---

## [Referee Report · Reviewer #2 (Public review)]

Summary

Bigge and colleagues use a sophisticated free-flight setup to study visuo-motor responses elicited in different parts of the visual field in the hummingbird hawkmoth. Hawkmoths have been previously shown to rely on translational optic flow information for flight control exclusively in the ventral and lateral parts of their visual field. Dorsally presented patterns, elicit a formerly completely unknown response - instead of using dorsal patterns to maintain straight flight paths, hawkmoths fly, more often, in a direction aligned with the main axis of the pattern presented (Bigge et al, 2021). Here, the authors go further and put ventral/lateral and dorsal visual cues into conflict. They found that the different visuomotor pathways act in parallel, and they identified a 'hierarchy': the avoidance of dorsal patterns had the strongest weight and optic flow-based speed regulation the lowest weight. The authors linked their behavioral results to visual scene statistics in the hawkmoths' natural environment. The partition of ventral and dorsal visuomotor pathways is well in line with differences in visual cue frequencies. The response hierarchy, however, seems to be dominated by dorsal features, that are less frequent, but presumably highly relevant for the animals' flight safety.

Strengths

The data are very interesting and unique. The manuscript provides a thorough analysis of free-flight behavior in a non-model organism that is extremely interesting for comparative reasons (and on its own). These data are both difficult to obtain and very valuable to the field.

Weaknesses

While the present manuscript clearly goes beyond Bigge et al, 2021, the advance could have perhaps been even stronger with a more fine-grained investigation of the visual responses in the dorsal visual field. Do hawkmoths, for example, show optomotor responses to rotational optic flow in the dorsal visual field?

I find the majority of the data, which are also the data supporting the main claims of the paper, compelling. However, the measurements of flight height are less solid than the rest and I think these data should be interpreted more carefully.

---

## [Referee Report · Reviewer #3 (Public review)]

The authors have significantly improved the paper in revising to make its contributions distinct from their prior paper. They have also responded to my concerns about quantification and parameter dependency of the integration conclusion. While I think there is still more that could be done in this capacity, especially in terms of the temporal statistics and quantification of the conflict responses, they have a made a case for the conclusions as stated. The paper still stands as an important paper with solid evidence a bit limited by these concerns.

[Editors' note: Due to very minor revisions, the paper was not sent to reviewers for an additional round of review.]

---

## [Author Response]

The following is the authors’ response to the previous reviews.

**Public Reviews:**

**Reviewer #1 (Public review):**
Summary:Recent work has demonstrated that the hummingbird hawkmoth, Macroglossum stellatarum, like many other flying insects, use ventrolateral optic flow cues for flight control. However, unlike other flying insects, the same stimulus presented in the dorsal visual field, elicits a directional response. Bigge et al., use behavioral flight experiments to set these two pathways in conflict in order to understand whether these two pathways (ventrolateral and dorsal) work together to direct flight and if so, how. The authors characterize the visual environment (the amount of contrast and translational optic flow) of the hawkmoth and find that different regions of the visual field are matched to relevant visual cues in their natural environment and that the integration of the two pathways reflects a prioritization for generating behavior that supports hawkmoth safety rather than the prevalence for a particular visual cue that is more prevalent in the environment.Strengths:This study creatively utilizes previous findings that the hawkmoth partitions their visual field as a way to examine parallel processing. The behavioral assay is well-established and the authors take the extra steps to characterize the visual ecology of the hawkmoth habitat to draw exciting conclusions about the hierarchy of each pathway as it contributes to flight control.
**Reviewer #2 (Public review):**
SummaryBigge and colleagues use a sophisticated free-flight setup to study visuo-motor responses elicited in different parts of the visual field in the hummingbird hawkmoth. Hawkmoths have been previously shown to rely on translational optic flow information for flight control exclusively in the ventral and lateral parts of their visual field. Dorsally presented patterns, elicit a formerly completely unknown response - instead of using dorsal patterns to maintain straight flight paths, hawkmoths fly, more often, in a direction aligned with the main axis of the pattern presented (Bigge et al, 2021). Here, the authors go further and put ventral/lateral and dorsal visual cues into conflict. They found that the different visuomotor pathways act in parallel, and they identified a 'hierarchy': the avoidance of dorsal patterns had the strongest weight and optic flow-based speed regulation the lowest weight. The authors linked their behavioral results to visual scene statistics in the hawkmoths' natural environment. The partition of ventral and dorsal visuomotor pathways is well in line with differences in visual cue frequencies. The response hierarchy, however, seems to be dominated by dorsal features, that are less frequent, but presumably highly relevant for the animals' flight safety.StrengthsThe data are very interesting and unique. The manuscript provides a thorough analysis of free-flight behavior in a non-model organism that is extremely interesting for comparative reasons (and on its own). These data are both difficult to obtain and very valuable to the field.WeaknessesWhile the present manuscript clearly goes beyond Bigge et al, 2021, the advance could have perhaps been even stronger with a more fine-grained investigation of the visual responses in the dorsal visual field. Do hawkmoths, for example, show optomotor responses to rotational optic flow in the dorsal visual field?I find the majority of the data, which are also the data supporting the main claims of the paper, compelling. However, the measurements of flight height are less solid than the rest and I think these data should be interpreted more carefully.
**Reviewer #3 (Public review):**
The authors have significantly improved the paper in revising to make its contributions distinct from their prior paper. They have also responded to my concerns about quantification and parameter dependency of the integration conclusion. While I think there is still more that could be done in this capacity, especially in terms of the temporal statistics and quantification of the conflict responses, they have a made a case for the conclusions as stated. The paper still stands as an important paper with solid evidence a bit limited by these concerns.
**Recommendations for the authors:**

**Reviewer #1 (Recommendations for the authors):**
The edits have significantly improved the clarity of the manuscript. A few small notes:Figure 2B legend - describe what the orange dashed line represents

We added a description.

Figure 2B legend - references Table 1 but I believe this should reference Table S1. There are other places in the manuscript where Table 1 is referenced and it should reference S1

We changed this for all instances in the main paper and supplement, where the reference was wrong.

Figure S1 legend - some figure panel letters are in parentheses while others are not

We unified the notation to not use parentheses for any of the panel letters.

**Reviewer #2 (Recommendations for the authors):**
I couldn't find the l, r, d, v indications in Fig. 1a. This was just a suggestion, but since you wrote you added them, I was wondering if this is the old figure version.

We added them to what is now Fig. 2, which was originally part of Fig. 1. After restructuring, we did indeed not add an additional set to Fig. 1, which we have now adjusted.

Fig. 2: Adding 'optic flow' and 'edges' to the y-axis in panels E and F, would make it faster for me to parse the figure. Maybe also add the units for the magnitudes? Same for Figure 6B

We added 'optic flow' and 'edges' to the panels E and F in Fig. 2 and Fig. 6.

Fig. 2: Very minor - could you use the same pictograms in D and E&F (i.e. all circles for example, instead of switching to "tunnels" in EF)?

We used the tunnel pictograms, because we associated those with the short notations for the different conditions summarised in Table S1. Because we wanted to keep this consistent across the paper, we used the “tunnel” pictograms here too.

In the manuscript, you still draw lots of conclusions based on these area measurements (L132-142, L204-209 etc). This does not fully reflect what you wrote in your reply to the reviewers. If you think of these measurements as qualitative rather than quantitative, I would say so in the manuscript and not use quantitative statistics etc. My suggestion would be to be more specific about potential issues that can influence the measurement (you mentioned body size, image contrast, motion blur, pitch across conditions etc) and give that data not the same weight as the rest of the measurements.

We do express explicit caution with this measure in the methods section (l. 657-659) and the results section (l. 135-137). Nevertheless, as the trends in the data are consistent with optic flow responses in the other planes, and with responses reported in the literature, we felt that it is valuable to report the data, as well as the statistics for all readers, who can – given out cautionary statement – assess the data accordingly.

The area measurements suggest that moths fly lower with unilateral vertical gratings (Fig. S1, G1 and G2 versus the rest). If you leave the data in can you speculate why that would be? (Sorry if I missed that)

We agree, this seems quite consistent, but we do not have a good explanation for this observation. It would certainly require some additional experiments and variable conditions to understand what causes this phenomenon.

Fig.4 - is panel B somehow flipped? Shouldn't the flight paths start out further away from the grating and then be moved closer to midline (as in A). That plot shows the opposite.

Absolutely right, thank you for spotting this, it was indeed an intermediate and not the final figure which was uploaded to the manuscript. It also had outdated letter-number identifiers, which we now updated.

L198 - should be "they avoided"

Corrected.